# Prevalence of *Giardia intestinalis* Infection in Schistosomiasis-Endemic Areas in South-Central Mali

**DOI:** 10.3390/tropicalmed4020086

**Published:** 2019-05-23

**Authors:** Hassan K.M. Fofana, Maren Schwarzkopf, Mama N. Doumbia, Rénion Saye, Anna Nimmesgern, Aly Landouré, Mamadou S. Traoré, Pascal Mertens, Jürg Utzinger, Moussa Sacko, Sören L. Becker

**Affiliations:** 1Institut National de Recherche en Santé Publique, B.P. 1771, Bamako, Mali; hkm459iii@gmail.com (H.K.M.F.); manieldk@gmail.com (M.N.D.); srenion@yahoo.fr (R.S.); aland954@hotmail.fr (A.L.); traorem@afribonemali.net (M.S.T.); msacko@afribonemali.net (M.S.); 2Institute of Medical Microbiology and Hygiene, Saarland University, 66421 Homburg/Saar, Germany; maren.schwarzkopf@gmail.com (M.S.); anna.nimmesgern@uks.eu (A.N.); 3Coris BioConcept, 5032 Gembloux, Belgium; pascal.mertens@corisbio.com; 4Swiss Tropical and Public Health Institute, P.O. Box, CH–4002 Basel, Switzerland; juerg.utzinger@swisstph.ch; 5University of Basel, P.O. Box, CH–4003 Basel, Switzerland

**Keywords:** BD Max Enteric Parasite Panel, diarrhea, *Giardia intestinalis*, Mali, polymerase chain reaction, rapid diagnostic test, *Schistosoma mansoni*, stool microscopy

## Abstract

Intestinal parasite infections are frequent causes of diarrhea and malnutrition among children in the tropics. Transmission of helminths and intestinal protozoa is intimately connected with conditions of poverty, including inadequate sanitation and hygiene. Concurrent infections with several intestinal pathogens may lead to excess morbidity. Yet, there is a paucity of epidemiological data from Mali. In this study, stool samples from 56 individuals, aged 2–63 years, from Bamako and Niono, south-central Mali were examined for intestinal parasites using stool microscopy. Additionally, stool samples were subjected to a rapid diagnostic test (RDT) and polymerase chain reaction (PCR) for the detection of *Cryptosporidium* spp. and *Giardia intestinalis*. The predominant pathogens were *Schistosoma mansoni* and *G. intestinalis* with prevalences of 41% and 38%, respectively. *Hymenolepis nana* was detected in 4% of the participants, while no eggs of soil-transmitted helminths were found. Concurrent infections with *G. intestinalis* and *S. mansoni* were diagnosed in 16% of the participants. For the detection of *G. intestinalis*, PCR was more sensitive (100%) than RDT (62%) and microscopy (48%). As helminth-protozoa coinfections might have important implications for morbidity control programs, future studies should employ diagnostic tools beyond stool microscopy to accurately assess the co-endemicity of giardiasis and schistosomiasis.

## 1. Introduction

Parasitic infections caused by helminths and intestinal protozoa are important causes of acute, persistent, and chronic digestive disorders in the tropics [1,2]. Parasitic infections are a public health problem in settings characterized by poor sanitation, inadequate access to clean water, and unhygienic behavior [3,4]. Infections are most commonly diagnosed by visualization of the causative agents (e.g., cysts, eggs, larvae, and trophozoites) on stool microscopy, a simple technique that is frequently available in resource-constrained settings [1,5]. However, its sensitivity is limited and epidemiological studies might thus underestimate the ‘true’ prevalence unless additional, more accurate techniques like antigen detection assays or polymerase chain reaction (PCR) are employed [6,7].

In Mali, schistosomiasis caused by *Schistosoma mansoni* and *Schistosoma haematobium* gives rise to considerable morbidity [8,9,10]. Morbidity might be further accentuated in the face of concurrent parasitic infections. While studies from southern Africa observed high rates of coinfection with schistosomiasis and giardiasis among children living in impoverished settings [11], data on *Giardia intestinalis* (synonymous: *G. duodenalis* and *G. lamblia*) infections in Mali are scarce. We carried out a study to assess the prevalence of *G. intestinalis* in schistosomiasis-endemic areas of Mali, using a suite of diagnostic approaches, including microscopy, antigen detection, and PCR.

## 2. Materials and Methods

### 2.1. Ethics Statement

This investigation was embedded in a clinical study on the occurrence of intestinal *Clostridium difficile* infections in Mali. The study protocol was approved by the ethics committee of the ‘Institut National de Recherche en Santé Publique’ (INRSP) in Bamako, Mali (approval no. 04/2016/CE-INRSP). Written informed consent was obtained from all participants before enrolment in the study. Parents or guardians signed on behalf of individuals younger than 18 years.

### 2.2. Study Area and Population

In March 2016, stool samples from individuals aged 2–63 years with self-reported gastrointestinal complaints (e.g., abdominal pain, abdominal bloating, or diarrhea during the preceding two weeks) were obtained in Bamako, the country’s capital, and Niono, a regional center located in a vast irrigation area known as ‘Office du Niger’, some 300 km north-east of Bamako [2]. Both Bamako and Niono are endemic for schistosomiasis [12].

### 2.3. Field and Laboratory Procedures

One fresh stool specimen was collected from each participant. All fecal specimens were subjected on the same day to stool microscopy using a direct fecal smear technique [13]. Microscope slides were read and analyzed by experienced laboratory technicians. Samples were stored at INRSP (Bamako, Mali) at 4 °C in a fridge for three weeks, before transfer to the Institute of Medical Microbiology and Hygiene at Saarland University (Homburg, Germany).

In Homburg, two further tests were employed, namely a rapid diagnostic test (RDT) for the concurrent detection of antigens shed by *Cryptosporidium* spp. and *G. intestinalis* (Crypto/Giardia DuoStrip, Coris BioConcept; Gembloux, Belgium) and a multiplex PCR assay, which detects *Cryptosporidium* spp., *Entamoeba histolytica*, and *G. intestinalis* (BD Max Enteric Parasite Panel, Becton Dickinson; Heidelberg, Germany). All tests were carried out as previously described [14,15]. In brief, the immunochromatographic RDT was performed as follows: approximately 10 μL of each stool specimen was taken with a loop and mixed with a buffer solution. Next, the RDT dipstick was put into this liquid, and results were read after exactly 15 min. Specific test bands indicate the presence or absence of *Cryptosporidium* spp. and *G. intestinalis*. For PCR examinations, the nucleic acid extraction and subsequent real-time PCR were performed on the commercially available BD Max™ instrument. A sample processing control was included in each run and all examinations were carried out by the same experienced laboratory technician who was blinded to the results of RDT and stool microscopy.

### 2.4. Statistical Analysis

Data were double-entered into an electronic database using Microsoft Excel. Any positive test result obtained by either PCR or stool microscopy was considered as ‘true’ positive for giardiasis (composite reference standard). Sensitivity, specificity, positive predictive value (PPV), and negative predictive value (NPV) were calculated for the individual diagnostic techniques.

## 3. Results

A total of 56 stool samples from individuals with self-reported gastrointestinal complaints in the two preceding weeks were examined, with 43 samples obtained from Niono and 13 specimens from Bamako. There were more female than male participants (33 vs. 23). Most participants (40; 71%) were children aged 6–15 years. The remaining participants were either preschoolers aged 5 years and below (*n* = 10; 18%) or adolescents and adults aged above 15 years (*n* = 6; 11%).

The prevalence of *S. mansoni* and *G. intestinalis* infections in our study sample was 41% and 38%, respectively. Coinfections with both parasites were observed in nine individuals (16%). As shown in Table 1, the helminth *Hymenolepis nana* and four species of non-pathogenic intestinal protozoa were also found, while neither *Cryptosporidium* spp. nor *E. histolytica* were detected.

All 21 *G. intestinalis* infections were correctly identified by PCR, whereas stool microscopy missed 11 of these infections. The sensitivity of microscopy, RDT, and PCR were 48%, 62%, and 100%, respectively (Table 2). Two ‘false’ positive RDT results were recorded, owing to a lower specificity of the RDT than that of microscopy and PCR (94% vs. 100%).

## 4. Discussion

We observed high prevalences of infection with *S. mansoni* and *G. intestinalis* in the study population in south-central Mali. Considering that only a single stool sample was examined from each participant, it is conceivable that the numbers reported here are underestimations of the ‘true’ prevalence. This remark is particularly true for helminth infections, which were only subjected to a relatively insensitive direct fecal smear. In contrast, diagnosis of intestinal protozoa was based on microscopy, supplemented with an RDT and PCR. While the BD Max™ PCR assay for intestinal protozoa has been validated in non-endemic settings [16,17], this study is—to our knowledge—the first to assess this molecular test on stool samples originating from West Africa. Eight *G. intestinalis* infections were exclusively detected on PCR, thus confirming the added value of molecular diagnostic tools. Exclusively PCR-positive samples might be characterized by a relatively low infection intensity, which may result in less clinical symptomatology, but plays an important role for continued transmission [18].

The co-endemicity of *G. intestinalis* and *Schistosoma* spp. infection has previously been identified as an insufficiently addressed public health issue in sub-Saharan Africa [11]. Indeed, while both infections are associated with water exposure, the specific infection routes differ. For acquiring *Schistosoma* infection, direct contact between an individual’s skin and freshwater containing infective cercariae is required (e.g., while bathing in a river or lake) [19]. Giardiasis is mainly acquired through oral ingestion of infective cysts (e.g., through drinking of contaminated water or recreational water exposure) [20,21]. Yet, the clinical significance of coinfections needs to be assessed in more detail. In our study, coinfection occurred in 16% of the participants and was not more common than expected by chance (15%). However, it is conceivable that the employed diagnostic approach missed additional cases of schistosomiasis, particularly those with low-intensity infection.

Our study is limited by the small and biased sample, which is not representative of the entire study population as we collected stool specimens from individuals with self-reported gastrointestinal symptoms only. We only examined one stool sample per participant, which might have resulted in an underestimation of the actual prevalence. Moreover, direct fecal smear, the technique used for stool microscopy, has only limited sensitivity for the detection of helminths and intestinal protozoa, and the RDT and PCR tests were carried out after several weeks of sample storage in a fridge, which might have negatively impacted the diagnostic accuracy of these two tests. Indeed, it cannot be ignored that the two ‘false’ positive RDT results observed in our study were ‘true’ positives, which might have gone undetected by PCR due to the degradation of the nucleic acids present in the respective samples. Future studies should thus be carried out on a larger sample and should employ more sensitive microscopic methods such as the Kato–Katz technique, formalin-ether concentration, or Mini-FLOTAC [1], which should ideally be complemented by molecular detection tests (e.g., PCR for *Schistosoma* spp., urine-based RDT detecting *Schistosoma*-specific circulating cathodic antigen (CCA), among other tests). Additionally, such studies should assess the infection intensity in individuals with *Giardia-Schistosoma* coinfections as compared to single infections. Finally, molecular testing for in-depth appraisal of helminths and intestinal protozoa would provide a more accurate profile of the prevailing gastrointestinal parasite infections in this setting.

## 5. Conclusions

We observed a substantial rate of giardiasis in schistosomiasis-endemic areas of south-central Mali, which warrants further investigation. The sole application of stool microscopy would have missed half of the *G. intestinalis* infections. The employed RDT was more sensitive than stool microscopy and might be suitable for rapid detection in resource-limited primary healthcare centers where PCR tests are not available.

## Figures and Tables

**Table 1 tropicalmed-04-00086-t001:** Prevalence of intestinal parasite infections in a study carried out among 56 individuals in Bamako and Niono, south-central Mali, in mid-2016.

Parasite	Prevalence		Age (Years)			Sex	
	N	%	≤5 (*n* = 10)	6–15 (*n* = 40)	>15 (*n* = 6)	Female (*n* = 33)	Male (*n* = 23)
*Schistosoma mansoni*	23	41	1	20	2	13	10
*Giardia intestinalis*	21	38	5	15	1	13	8
*Entamoeba coli*	10	18	2	6	2	5	5
*Hymenolepis nana*	2	4	0	2	0	0	2
*Chilomastix mesnili*	2	4	0	2	0	2	0
*Entamoeba dispar* ^a^	2	4	0	2	0	1	1
*Trichomonas intestinalis*	1	2	0	1	0	1	0

^a^*E. dispar* and *E. histolytica* cannot be distinguished on stool microscopy, but the two microscopically positive samples tested negative for *E. histolytica* on PCR.

**Table 2 tropicalmed-04-00086-t002:** Comparison of the diagnostic accuracy of different laboratory tests for the detection of *Giardia intestinalis* infection in the same study. Any positive test result obtained by either PCR or stool microscopy was considered as ‘true’ positive for *G. intestinalis* infection (composite reference standard).

Test for *G. intestinalis*	Positivity Rate		Sensitivity	Specificity	PPV	NPV
	N	%				
Stool microscopy	10	18	48%	100%	100%	76%
RDT	15	27	62%	94%	87%	81%
PCR	21	38	100%	100%	100%	100%

NPV, negative predictive value; PCR, polymerase chain reaction PPV, positive predictive value; RDT, rapid diagnostic test.

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
