# Peer review of "Prevalence of Giardia intestinalis Infection in Schistosomiasis-Endemic Areas in South-Central Mali"

_tropicalmed, 2019, doi:10.3390/tropicalmed4020086_

Round 1
Reviewer 1 Report
Authors described an intestinal parasite investigation in people in Schistosomiasis endemic areas in Mali and identified that people with gastrointestinal disturbance had high prevalence of Giardia intestinalis and Schistosoma mansoni infections, both water-borne parasites. Authors also found that DNA detection in fecal sample using PCR was more sensitive to detect Giardia infections than traditional wet mount microscopy and antigen detection with RDT. The finding is informative to interpret the co-infection of Giardiasis and schistosomiasis and the impact on the pathology, and the need of the more sensitive methods to detect intestinal protozoan infection. Some suggestions are given as following.
1. I suggest changing the species name of Giardia intestinalis to Giardia lamblia, the latter is more formal and popularly used.
2. Except for the co-infection described in this study, has the people with S. mansoni infection higher infection rate of G. lamblia? This information would interpret if infection of Schistosome reduces the intestinal defense and increases the susceptible to other intestinal protozoa infections, or vise verse.
3. Simultaneous multiple PCR detection of fecal sample for other gastrointestinal parasites, except for Giardia, may give a more accurate profile of gastrointestinal parasite infections in these population.
Author Response
Authors described an intestinal parasite investigation in people in Schistosomiasis endemic areas in Mali and identified that people with gastrointestinal disturbance had high prevalence of Giardia intestinalis and Schistosoma mansoni infections, both water-borne parasites. Authors also found that DNA detection in fecal sample using PCR was more sensitive to detect Giardia infections than traditional wet mount microscopy and antigen detection with RDT. The finding is informative to interpret the co-infection of Giardiasis and schistosomiasis and the impact on the pathology, and the need of the more sensitive methods to detect intestinal protozoan infection. Some suggestions are given as following.
Response: We are indebted to Reviewer #1 for having studied our manuscript in such a timely manner and for having provided a set of specific comments and suggestions.
1. I suggest changing the species name of Giardia intestinalis to Giardia lamblia, the latter is more formal and popularly used.
Response: We agree that the three terms G. intestinalis, G. lamblia and G. duodenalis have been used interchangeably in the scientific literature. Hence, we have inserted the following statement in the Introduction of our manuscript: “[…] data on Giardia intestinalis (synonymous: G. duodenalis and G. lamblia) infections in Mali are scarce” (see revised manuscript, lines 51-52). Our suggestion is to keep the term “Giardia intestinalis” in the title and throughout the manuscript, as there is growing consensus among clinicians, microbiologists and public health experts to use this species name.
2. Except for the co-infection described in this study, has the people with S. mansoni infection higher infection rate of G. lamblia? This information would interpret if infection of Schistosome reduces the intestinal defense and increases the susceptible to other intestinal protozoa infections, or vise verse.
Response: Coinfections with schistosomiasis and giardiasis were not more common than expected by chance, as described in the Discussion: “In our study, coinfection occurred in 16% of the participants and was not more common than expected by chance. However, it is conceivable that the employed diagnostic approach missed additional cases of schistosomiasis, particularly those with low-intensity infection” (see revised manuscript, lines 138-140). The infection intensity of intestinal protozoa species was not rigorously assessed, and we have added this as a study limitation: “Additionally, such [future] studies should assess the infection intensity in individuals with Giardia-Schistosoma coinfections as compared to single infections“ (see revised manuscript, lines 154-156).
3. Simultaneous multiple PCR detection of fecal sample for other gastrointestinal parasites, except for Giardia, may give a more accurate profile of gastrointestinal parasite infections in these population.
Response: Additional PCR analyses for other intestinal pathogens were not feasible in this study. However, we provide a statement pertaining to a more detailed characterisation of intestinal pathogens in the Discussion of our revised manuscript: “Finally, molecular testing for in-depth appraisal of helminths and protozoa would provide a more accurate profile of the prevailing gastrointestinal parasite infections in this setting” (see revised manuscript, lines 156-158).
Reviewer 2 Report
The manuscript about detection of Giardia intestinalis infection in an endemic area of Schistosomiasis, by comparison of three different diagnostic test results, although probably is not very original, has interest for the readers. However, I feel that the present manuscript could be considered as a research note instead of a full- article. A future study with more samples, together with the employment of more sensitive microscopy methods, will offer more interesting results and conclusions.
In any case, I have several comments, which could improve the interpretation of the results:
- The title could be changed erasing the qualifying adjective "High"
- How the experienced laboratory technicians were able to identify Trichomonas intestinalis and could not diagnosis 11 cases of Giardia??? It is much more difficult to find the trophozoit of T. intestinalis than the cysts or trophozoit of Giardia.
- In Table 2 data are confused with respect to RDT prevalence calculations. It could be better detail the two false positives, resulting in 13 infections (23% prevalence and 62% sensitivity)
- Table 2 results seem to be incorrect: the specificity of RDT seems to be 62% instead of 94% written in Table 2. How do the authors calculate these values?
- Again in Table 2, although for microscopy and PCR calculations of PPV and NPV are correct, those from RDT are incorrect: PPV seems to be 68% and NPV 85%, instead of 87% and 81%, respectively. How do the authors calculate these values?
- line 65: …..”diarrhoea during the preceding 2 weeks”. That means that you only take diarrheic samples or that the gastrointestinal complaints were already disappeared?? A short explanation can be included in the text
- line 73: …..”at 4ºC in the fridge for 3 weeks”. That means without any fixative??
- line 80: ….”Approximately 10ul”. Were all the sample diarrheic?? It could be better use ugr for stool samples, with a sentence like: “xxx ugr were taken and mixed with 10 ul buffer solution”
- line 95: the authors give the number of samples obtained in both localities. However why they do not calculate any differences between the capital and Niono?
- line 111: It could be necessary to explain that the PCR is considered as the reference diagnostic test with the high sensitivity (100%)
- line 126: “Eight out of 21 G. intestinalis infections were exclusively detected by PCR”. This is not true, because with RDT authors already detect 15 infections, so with PCR they exclusively detect 6 more G.intestinalis infections (21-15=6). (However, if authors correct the values in Table 2, with 13 infections detected by RDT, it would be correct that 8 infections were exclusively detect by PCR)
Author Response
The manuscript about detection of Giardia intestinalis infection in an endemic area of schistosomiasis, by comparison of three different diagnostic test results, although probably is not very original, has interest for the readers. However, I feel that the present manuscript could be considered as a research note instead of a full- article. A future study with more samples, together with the employment of more sensitive microscopy methods, will offer more interesting results and conclusions.
Response: We thank Reviewer #2 for the generous appraisal of our piece and for offering a series of specific suggestions. We agree that our findings will need to be confirmed in future studies with a larger sample size. However, as this study is among the first investigations from West Africa that focus on the association of giardiasis and schistosomiasis, it would be our pleasure to keep the manuscript in its current form, i.e. as full article.
In any case, I have several comments, which could improve the interpretation of the results:
- The title could be changed erasing the qualifying adjective "High"
Response: We have followed this suggestion and omitted the adjective “high” from the title of our manuscript (see revised manuscript, line 2).
- How the experienced laboratory technicians were able to identify Trichomonas intestinalis and could not diagnosis 11 cases of Giardia?? It is much more difficult to find the trophozoit of T. intestinalis than the cysts or trophozoit of Giardia.
Response: We agree that the identification of Trichomonas intestinalis on stool microscopy is challenging. However, we hypothesise that the 11 microscopy-negative Giardia samples can rather be explained by low quantities of G. intestinalis being present in the examined sample. Indeed, most studies from high- and low-income countries comparing stool microscopy to molecular tests for the diagnosis of giardiasis report a considerable number of microscopy-negative, but PCR-positive samples for G. intestinalis.
- In Table 2 data are confused with respect to RDT prevalence calculations. It could be better detail the two false positives, resulting in 13 infections (23% prevalence and 62% sensitivity)
Response: We have slightly modified Table 2 to reflect Reviewer #2’s suggestion. Indeed, the term “positivity rate” of the individual diagnostic tests for G. intestinalis detection is more accurate. Hence, we have replaced the term “prevalence” with “positivity rate” (see revised manuscript, Table 2).
- Table 2 results seem to be incorrect: the specificity of RDT seems to be 62% instead of 94% written in Table 2. How do the authors calculate these values?
Response: We have carefully checked the data. Please note that the rate of ‘true’ negatives correctly identified as such by the rapid diagnostic test (RDT) is 33 out of 35 (since there were 2 ‘false’ positive samples in the RDT); and hence 33/35 = 0.94, which equals a specificity of 94%. Hence, no corrective action was performed.
- Again in Table 2, although for microscopy and PCR calculations of PPV and NPV are correct, those from RDT are incorrect: PPV seems to be 68% and NPV 85%, instead of 87% and 81%, respectively. How do the authors calculate these values?
Response: We have double-checked the calculations. PPV is defined as the number of ‘true’ positives divided by the number of positive calls in a test (e.g. for the RDT in our study: 13/(13+2) = 0.87 à 87%). Likewise, NPV is defined as the number of ‘true’ negatives divided by the sum of ‘true’ and ‘false’ negatives. For the RDT used in our study, this translates into 35/43 = 0.81 à 81%. Hence, no corrective actions were deemed necessary. However, to enhance the comprehensibility of Table 2, we have added the definition of our diagnostic composite reference standard, which is also mentioned in the Methods, in the legend of Table 2 (i.e. “Any positive test result obtained by either PCR or stool microscopy was considered as ‘true’ positive for G. intestinalis infection (composite reference standard))” (see revised manuscript, lines 113-115).
- line 65: …..”diarrhoea during the preceding 2 weeks”. That means that you only take diarrheic samples or that the gastrointestinal complaints were already disappeared? A short explanation can be included in the text
Response: Self-reported morbidity was employed to assess eligibility for inclusion in the current study. Hence, patients were invited to participate if they had suffered from symptoms like either diarrhoea or abdominal pain or bloating within the 2 preceding weeks. We have detailed this in the text (see revised manuscript, lines 63-65).
- line 73: …..”at 4ºC in the fridge for 3 weeks”. That means without any fixative?
Response: We confirm that no fixative was used to preserve the stool samples. This is clearly mentioned as limitation in our revised piece (see revised manuscript, lines 146-150). However, it should be noted that previous studies have shown that the PCR test used in our study accurately detects intestinal protozoa infections even after up to 10 months of storage at disrupted cold chain conditions without the use of a fixative (see revised manuscript, reference no. 15).
- line 80: ….”Approximately 10ul”. Were all the sample diarrheic?? It could be better use ugr for stool samples, with a sentence like: “xxx ugr were taken and mixed with 10 ul buffer solution”
Response: Not all stool samples in the study were diarrhoeic, but a sampling loop designed for a quantity of 10 μl was used to process all the samples, regardless of their individual characteristics. Hence, we would like to keep the current wording.
- line 95: the authors give the number of samples obtained in both localities. However why they do not calculate any differences between the capital and Niono?
Response: We thank Reviewer #2 for this observation. As acknowledged above, the sample size in our study is limited (n=56) and we thus believe that our results are more meaningful if they are presented for the entire cohort, as the groups might otherwise become too small for useful and accurate calculations of the diagnostic accuracy of the various tests used for detection of G. intestinalis infection.
- line 111: It could be necessary to explain that the PCR is considered as the reference diagnostic test with the high sensitivity (100%)
Response: We agree with Reviewer #2 and have added this explanation in the legend of Table 2 (see revised manuscript, lines 113-115).
line 126: “Eight out of 21 G. intestinalis infections were exclusively detected by PCR”. This is not true, because with RDT authors already detect 15 infections, so with PCR they exclusively detect 6 more G.intestinalis infections (21-15=6). (However, if authors correct the values in Table 2, with 13 infections detected by RDT, it would be correct that 8 infections were exclusively detect by PCR)
Response: We thank Reviewer #2 for having examined our manuscript so carefully. As explained above, the RDT correctly identified 13 infections (and two ‘false’ positive samples); hence, the statement is correct. Following our aforementioned modifications in Table 2, this should now be more easily understandable when reading our manuscript. Additionally, we have slightly modified the concerned sentence in the Discussion, as follows: “Eight G. intestinalis infections were exclusively detected on PCR, thus confirming the added value of molecular diagnostic tools” (see revised manuscript, lines 127-128).